# A Wearable Flexible Acceleration Sensor for Monitoring Human Motion

**DOI:** 10.3390/bios12080620

**Published:** 2022-08-10

**Authors:** Zeqing He, Kuan Wang, Zhao Zhao, Taihua Zhang, Yuhang Li, Liu Wang

**Affiliations:** 1Aerospace Information Research Institute, Chinese Academy of Sciences, Beijing 100094, China; 2Institute of Solid Mechanics, Beihang University (BUAA), Beijing 100191, China; 3Key Laboratory of Biomechanics and Mechanobiology of Ministry of Education, Beijing Advanced Innovation Center for Biomedical Engineering, School of Biological Science and Medical Engineering, School of Engineering Medicine, Beihang University, Beijing 100083, China

**Keywords:** wearable electronics, flexible electronics, acceleration sensor

## Abstract

Skin-inspired flexible wearable acceleration sensors attract much attention due to their advantages of portability, personalized and comfortable experience, and potential application in healthcare monitoring, human–machine interfaces, artificial intelligence, and physical sports performance evaluation. This paper presents a flexible wearable acceleration sensor for monitoring human motion by introducing the island–bridge configuration and serpentine interconnects. Compared with traditional wearable accelerometers, the flexible accelerometer proposed in this paper improves the wearing comfort while reducing the cost of the device. Simulation and experiments under bending, stretching, and torsion conditions demonstrate that the flexible performance of the flexible acceleration sensor can meet the needs of monitoring the daily movement of the human body, and it can work normally under various conditions. The measurement accuracy of the flexible acceleration sensor is verified by comparing it with the data of the commercial acceleration sensor. The flexible acceleration sensor can measure the acceleration and the angular velocity of the human body with six degrees of freedom and recognize the gesture and motion features according to the acceleration characteristics. The presented flexible accelerometers provide great potential in recognizing the motion features that are critical for healthcare monitoring and physical sports performance evaluation.

## 1. Introduction

Skin-inspired flexible wearable signal sensors such as temperature [1,2,3], strain [4,5,6], pressure [7,8,9], acceleration [10,11,12,13] etc. attract much attention due to their advantages of portability, personalized and comfortable experience, and potential application in healthcare monitoring [14,15,16], human-machine interfaces [17,18], artificial intelligence [19], and physical sports performance evaluation [20,21]. The strain sensor designed by Zhang et al. [22] has been used to monitor the movement of human joints, but the joint movement cannot reflect the overall movement of the human body. Among diverse wearable sensors, acceleration sensors capable to monitor physical movement and motion play an essential role in health condition detection and clinical management, such as in Parkinson’s disease [23]. Strategies to reformulate conventional rigid devices into novel flexible wearable electronics include both materials and structural designs [24,25,26], which enable devices conformally attached to the soft and curvilinear surfaces of human skin without discomfort [27]. The first approach is introducing intrinsically flexible materials such as soft polymers [28,29,30,31], organogels [32,33], and hydrogels [31,34,35], which readily adapt to soft skin. Nevertheless, the only adoption of flexible materials cannot necessarily ensure the flexibility of the entire electronics. Yamamoto et al. [11], installed the three-axis acceleration sensor on the kirigami structure PET sheet. Although the kirigami structure improves the wearing comfort of the device, the stretchability of the device is not large. Kim et al. [36] studied the fabrication and deformation response of non-coplanar stretchable electronic devices. The circuit structure is tightly bound to the elastic substrate, resulting in a system with a reversible elastic response to extreme mechanical deformation. The design offers both excellent electrical performance and elastic deformation at high strain levels in different configurations. In order to further guarantee the flexible performance of the entire device, architectural design through introducing island–bridge configuration, serpentine, or buckling structures of the connections is adopted [37,38]. Furthermore, the location of the neutral mechanical plane in the device of multilayer structures plays a crucial role in the bending property of the device. To ensure better bending property, a layout design according to a multilayer neutral mechanical plane to define the position where strains are zero is preferred [39,40].

In this study, we design and fabricate a flexible wearable acceleration sensor for human motion detection by combining intrinsically flexible Ecoflex, island–bridge configuration and serpentine structure connections, which can be conformally attached to the human body. Experiments and mechanical analyses show that this flexible acceleration sensor can work in normal operation conditions without failure under bending, stretching, and twisting movements. In addition, the monitoring performance of the flexible acceleration sensor under vibration conditions is comparable to that of the commercial accelerometer. Furthermore, the flexible wearable acceleration sensors demonstrate excellent motion identification capability through acceleration and angular velocity detection.

## 2. Results and Discussion

The schematic exploded view illustration of the flexible acceleration sensor is shown in Figure 1a, and the two-dimensional schematic diagram of the flexible accelerometer is given in Figure 1b. The device (length, ~35 mm, width, ~35 mm) is composed of serpentine copper (Cu) conductor (thickness, ~18 μm, width, ~254 μm) on a polyimide (PI) substrate layer (thickness, ~30 μm, width, 654 μm) that used to connect major components including MPU6050, STM 32, capacitor, resistor, crystal oscillator and pin, PI insulation layer (thickness, 20 μm), and Ecoflex encapsulation layer (thickness, ~1 mm). Among them, STM32 is used to drive MPU6050 and read raw data; MPU6050 is a six-axis sensor chip used to measure motion data; crystal oscillator provides the clock required for communication for each part; and the pin is used to connect data line/power supply. In order to optimize the connection method and component arrangement to develop a miniaturized acceleration sensor, the bilayer structure of Cu wire is adopted. Figure 1c,d shows the photograph of the flexible wearable acceleration sensor and its conformal adhesion on the wrist.

The proposed flexible acceleration sensor can be directly attached to the human skin. In order to perfectly move with the human body, the flexible acceleration sensor needs to sustain complex deformations and their combinations such as bending, stretching or twisting. However, large internal strains in the copper wires, resulting from the deformations in the devices, may lead to the failure of the flexible acceleration sensor. To further demonstrate the flexibility and robustness of the device, relevant experiments and mechanical analyses of the device were carried out. In Figure 2, experiments on bending, stretching, and twisting deformations are shown and the device can work in normal operation conditions without failure. In order to illustrate the internal strains in the copper wires, the finite element model was conducted via commercial software ABAQUS. In the finite element method model, the shell element (S4R) is used for the Cu/PI lamination and the hexahedron elements (C3D8R) is used in the Ecoflex. The Young’s modulus of PI and Cu are 2.5 GPa and 119 GPa, and their corresponding Poisson’s ratios are 0.34 and 0.34. Ecoflex uses the Mooney–Rivlin hyperelastic constitutive model with parameters *C*_10_ = 0.048 MPa, *C*_01_ = −0.152 MPa [41]. Additionally, the mesh convergence was studied in the model. It can be seen that the deformations in the FEA model are very similar to those in the experiment results and the finite element model can provide the maximum strain in the copper wire. In Figure 2a, the FEA model shows that the maximum strain of the copper wire is 0.23% when the device is bent. Figure 2b shows the device is stretched with 10% external strain, and the FEA results illustrate that the maximum strain in copper wires can reach 1.2%. In Figure 2c, when the device is twisted, the maximum strain in copper wires is 0.27%. These experiments and simulations demonstrate the robust stability of the flexible accelerometer as a wearable device.

For daily regular movement, the selected STM32-driven MPU6050 chip is capable of both three axial accelerations and three axial angular velocities monitoring to attain attitude calculation and has sufficient measurement accuracy for the flexible acceleration sensor. However, there are other components inside the flexible acceleration sensor and the structure is relatively complex, which may interfere with the measurement results. To verify the measurement accuracy of the flexible acceleration sensor, a vibration test was carried out and compared with that of commercial accelerometers. Figure 3a,b demonstrate the experimental setting of the vibration test. The vibration signal, including the frequency and initial amplitude, can be set and generated by the arbitrary waveform generator, and the power amplifier can amplify the generated signal and then transmit it to the vibration exciter. The flexible acceleration sensor and commercial accelerometer adhere to the surface of the vibration platform. Figure 3c–e are the data under the vibration frequencies of 2 Hz, 4 Hz, and 6 Hz of Z direction, respectively. It can be seen from the figure that although the amplitude of the flexible acceleration sensor is slightly larger than that of the commercial acceleration sensor at 2 Hz, the frequency has no obvious difference. In addition, both the amplitude and frequency are in agreement at 4 Hz and 6 Hz. The small amplitude error that the flexible acceleration sensor may generate at 2 Hz is acceptable, as the application scenario of the device is to detect the motion of the human body and identify different motions through the six-degrees-of-freedom data of different motions.

To demonstrate the detection function of human motion, the flexible acceleration sensor is attached to the arm and real-time data from the acceleration sensor induced by human motion is collected. Figure 4 shows acceleration and angular velocity of/around *X*-axis, *Y*-axis, and *Z*-axis direction obtained during corresponding specific motion. *X*-axis and *Y*-axis acceleration of the sensor along with the direction of the thumb and the direction of the other fingers is positive, respectively, when the thumb is perpendicular to the other four fingers. The *Z*-axis acceleration of the sensor along with the upward direction which is perpendicular to the back of the hand is positive. For arm twisting motion, the acceleration of the *Y*-axis is larger than that of the *X*-axis, while the acceleration value of the *Z*-axis is almost zero. The period of the acceleration of the *Y*-axis is twice that of the *X*-axis. The angular velocity around the *Z*-axis is much larger than that of the other two directions. In addition, the phase of the angular velocity around the *X*-axis is almost similar compared to that around the *Z*-axis, but is opposite to that around the *Y*-axis, as shown in Figure 4a. Figure 4b shows the acceleration and angular velocity and the corresponding optical image of squatting motion. The maximum *Z*-axis acceleration and minimum *X*-axis acceleration are generated with the up-and-down swing of the arm, and a medium *Y*-axis acceleration is generated with the back-and-forth swing of the body. Figure 4c presents the results and the corresponding optical image of the jump motion, in which the arm maintains a natural drooping state. The acceleration of *Y*-axis is the largest, but that of the *X*-axis and *Z*-axis is almost zero. The slight angular velocities around the *X*-axis, *Y*-axis and *Z*-axis are produced by swinging of the arm. Figure 4d shows the results of the running motion. The change trend of the *X*-axis acceleration and the *Y*-axis acceleration is almost consistent, but the magnitude of the amplitude is different. The change in the trend of *Z*-axis acceleration is opposite to that of the *X*-axis, and the angular velocity around the *Z*-axis is the largest. The amplitude of the angular velocity around the *X*-axis is almost the same as that of the *Y*-axis, while the phase is opposite. Generally, these results reveal that unique data characteristics can be generated by different movement, which made the identification of specific movement through flexible acceleration sensor promising.

## 3. Conclusions

In conclusion, this paper proposes a flexible acceleration sensor through introducing both intrinsically flexible Ecoflex and serpentine interconnects, which can realize real-time specific motion recognition and monitoring by measuring three acceleration and three angular velocities. Experiments and mechanical analyses show that the flexible acceleration sensors can work in normal operation conditions without failure under bending, stretching, and twisting movements. In addition, the monitoring performance of the flexible acceleration sensor under vibration conditions is comparable to that of a commercial accelerometer. The flexible acceleration sensor presented in this paper provides a highly flexible and adaptable solution for healthcare monitoring and athletic performance assessment applications.

## 4. Materials and Methods

### 4.1. Fabrication Process of the Flexible Accelerometer

Flexible print circuit board (FPCB, three-layer laminate consisting of 18 μm copper and 30 μm PI) was cut into the desired shape using a programmable laser cutter (DCT, DL300U, Tianjin, China). In the next step, 1 mm of Ecoflex (A/B component 1:1 configuration) was spread on the bottom layer of the circular apparatus, then placed in a vacuum box to extract air bubbles, and then heated to 50° and cured for 4 min. We then laid the cut FPCB on the surface of the Ecoflex that was not fully cured, poured 1 mm of Ecoflex, and placed it at room temperature to cure after vacuuming. After curing, it was cut into 35 mm × 35 mm devices.

### 4.2. Mechanics Simulation

Mechanical simulation is performed using the commercial software ABAQUS. In the finite element method model, the shell element (S4R) is used for the Cu/PI lamination, and the hexahedron elements (C3D8R) is used in the Ecoflex. The Young’s modulus of PI and Cu are 2.5 GPa and 119 GPa, respectively, and their corresponding Poisson’s ratios are 0.34 and 0.34, respectively. Ecoflex uses the Mooney–Rivlin hyperelastic constitutive model with the parameters *C*_10_ = 0.048 MPa and *C*_01_ = −0.152 MPa. We simulated the adhesion process (device bending), biaxial stretching, and torsion scenarios, respectively, and determined that the principal strain distribution of the device under different scenarios.

### 4.3. Mechanical Measurements

We put the uncut device into a biaxial extensometer for isometric stretching during biaxial stretching. The biaxial extensometer consists of four commercial slides (HongYuanTai) and four 3D-printed grips (AURORA), with a precise measurement of 0.1 mm, capable of guaranteeing isometric stretching motions. According to the Saint Venant principle, the sample in the central region will be stretched uniformly and isotropically without being affected by boundary effects. Using a laser-displacement sensor (Micro-Epsilon, optoNCDT1420, Ortenburg, Germany), the deformation of the device was measured without touching the sample.

### 4.4. Vibration Test

First, we connected the arbitrary waveform generator (RIGOL, DG1022U, Beijing, China), power amplifier (NUAA, HEA-200C, Nanjing, China), vibration exciter (NUAA, HEV-200, Nanjing, China), flexible acceleration sensor, and USB to a serial module, the commercial acceleration sensor (PCB, 333B30, Anaheim, CA, USA), data acquisition instrument (NI, USB-4431, Austin, TX, USA), and computer according to Figure 3a,b. Then, the flexible and commercial accelerometer was adhered to the surface of the vibration platform. The arbitrary waveform generator generates the vibration signal. We then set the frequency and initial amplitude of the signal and transmitted the amplified signal to the vibration exciter through the power amplifier. The measurement data of the flexible acceleration sensor was read through the host computer software, and the measurement data of the commercial acceleration sensor was obtained through the data acquisition instrument and the commercial software LabVIEW.

## Figures and Tables

**Figure 1 biosensors-12-00620-f001:**
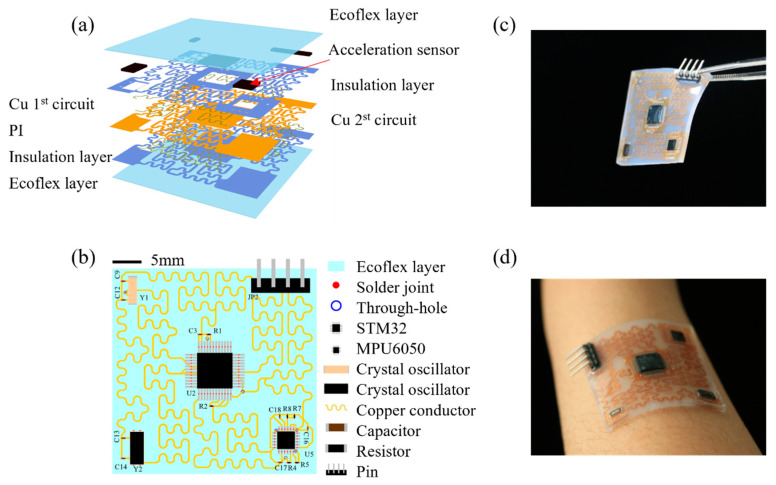
(**a**) Schematic exploded view illustration of the flexible acceleration sensor. (**b**) Two-dimensional schematic diagram of the flexible acceleration sensor. (**c**) Photograph of the flexible wearable acceleration sensor. (**d**) Photograph of the flexible wearable sensor attached to the human arm conformally.

**Figure 2 biosensors-12-00620-f002:**
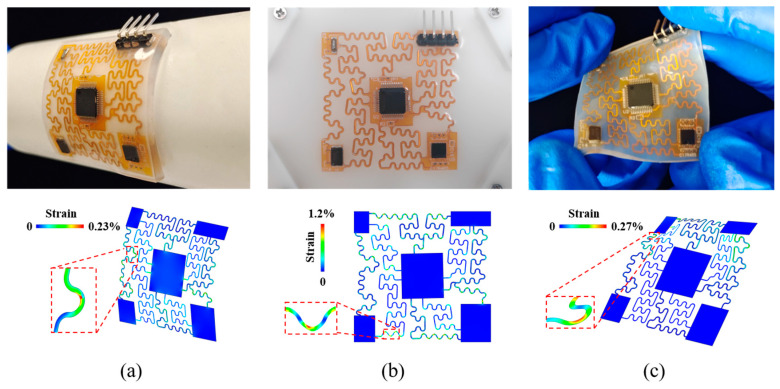
Three-dimensional morphology of the copper wire of the accelerometer and maximum principal strains obtained by experiments and mechanical simulations with external loading conditions of (**a**) bending, (**b**) stretching, (**c**) twisting.

**Figure 3 biosensors-12-00620-f003:**
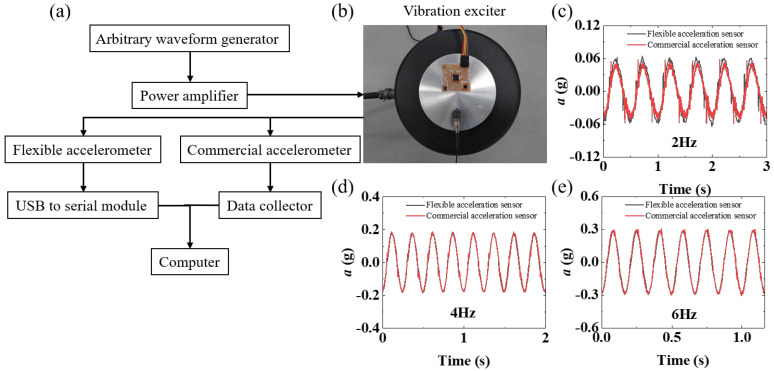
(**a**) Experimental setup for vibration tests, including arbitrary waveform generators, power amplifiers, flexible accelerometer, commercial accelerometer, USB to serial module, data collector, computer, and (**b**) vibration exciter. The Z−direction acceleration measurement data of the flexible accelerometer and commercial accelerometer at (**c**) 2 Hz, (**d**) 4 Hz, and (**e**) 6 Hz, respectively.

**Figure 4 biosensors-12-00620-f004:**
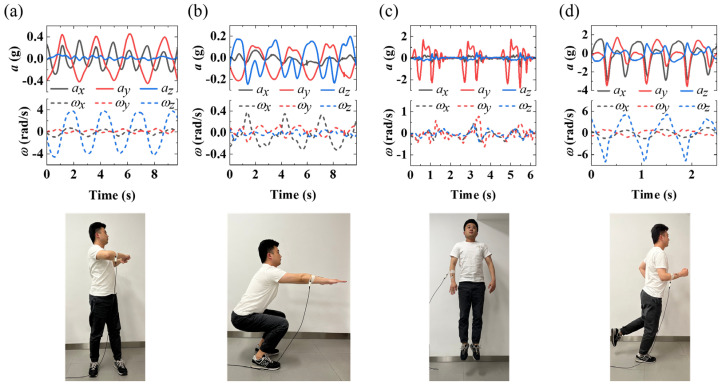
Real-time acceleration and axial angular of/around *X*-axis, *Y*-axis, and *Z*-axis direction during (**a**) arm twisting, (**b**) squatting, (**c**) jumping, (**d**) running.

## Data Availability

Not applicable.

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
