# Peer review of "A Wearable Flexible Acceleration Sensor for Monitoring Human Motion"

_biosensors, 2022, doi:10.3390/bios12080620_

Round 1

Reviewer 1 Report

Flexible wearable acceleration sensors attract much attention due to their  advantages of portability, personalized and comfortable experience, and potential application in healthcare monitoring, human-machine interfaces, AI, and evaluation of performance physical sports. 

The Authors presents a flexible wearable acceleration sensor for monitoring human  motion by introducing the island-bridge configuration and serpentine structure interconnects. They present proof-of-concept, FEM analysis and compare it with a commercial sensor.

The manuscript can be accepted for the publication in Biosensors after a minor revision.

- please read the manuscript and check all typos etc. In Introduction the literature is sometimes cited without brackets, see line 38, 42, 45.

- what is Ecoflex shell? where can it be bought or how to prepare it? please add some basic information.

- the proper electronic scheme of the proposed flexible acceleration sensor should be added to the manuscript

- could authors elaborate more on the proposed sensor in terms of other flexible acceleration sensors available on the market?

Reviewer 2 Report

This paper presents a wearable acceleration sensor with serpentine interconnects. They used 4 different islands to differentiate the rigid components. Then the entire system is encapsulated with ecoflex elastomer for further use for capturing various motions. Here are some minor questions:

1)      It seems that the power is externally connected. What is the requirement of the system to be integrated with batteries? What will be the obstacles?

2)      The main measurement component is acceleration sensor which is already rigid. Could authors explain the rationale for why the system should be stretchable rather than combining in a small device?

3)      In figure 4, how do authors think when the sensor is attached to different place of the body and the signal is changed? If so, any ideas to figure it out?
